# The Relationship between the Infertility Specialist and the Patient during the COVID-19 Pandemic

**DOI:** 10.3390/healthcare9121649

**Published:** 2021-11-28

**Authors:** Diana Antonia Iordăchescu, Florinda Tinella Golu, Corina Ioana Paica, Adrian Gorbănescu, Anca Maria Panaitescu, Corina Gică, Gheorghe Peltecu, Nicolae Gică

**Affiliations:** 1Faculty of Psychology and Educational Sciences, University of Bucharest, 050653 Bucharest, Romania; diana-antonia.iordachescu@fpse.unibuc.ro (D.A.I.); corina.paica@unibuc.ro (C.I.P.); adrian.gorbanescu@unibuc.ro (A.G.); 2Filantropia Clinical Hospital, Carol Davila University of Medicine and Pharmacy, 050474 Bucharest, Romania; anca.panaitescu@umfcd.ro (A.M.P.); mat.corina@gmail.com (C.G.); gheorghe.peltecu@umfcd.ro (G.P.); gica.nicolae@umfcd.ro (N.G.)

**Keywords:** doctor–patient relationship, explanatory model, communication, empathy, treatment compliance

## Abstract

The doctor–patient relationship is fundamental in the treatment of infertility, due to the emotional implications of fertilization procedures. However, insufficient data are available specifically for this relationship. The general objective of the study is to establish the associations between the fundamental concepts that define the doctor–patient relationship: communication, empathy, trust, collaboration, compliance and satisfaction. A cross-sectional study was conducted between May and June 2020 and followed the methods of a quantitative analysis, collecting the data using questionnaires. The research plan was specific to path analysis with the mediation effect, in which the hypotheses were tested. The research group consisted of 151 women diagnosed with infertility, voluntarily recruited through online support communities. Findings demonstrate that affective empathy mediates the relationship between communication and trust in the doctor. In conclusion, this study draws attention to the importance of basic concepts in the relationship of infertility specialists with infertile patients. Thus, it is necessary for health care providers in assisted human reproduction to participate in programs for the continuous training of empathic communication skills, given the sensitivity of this diagnosis.

## 1. Introduction

Over the years, infertility has been studied in terms of different approaches. Clinical and medical psychology aims to evaluate the impact of this medical condition on the mental health of infertile couples, while the biomedical approach has generated a large amount of pathophysiological information and guidelines for diagnostic and therapeutic interventions [1]. Although there has been consistent progress in both medical and psychosocial treatment, there are still gaps in the literature regarding the relationship between the infertility specialist and the patient diagnosed with infertility. The patient–doctor relationship is, therefore, important because this can help to offset some of the negative emotional experiences that occur during infertility treatments. In addition, the development of medical services involved in the treatment of infertility, as well as the promotion of information, understanding and awareness of the phenomenon, has led to an increase in treatment requests. Thus, it is necessary for the infertility specialist to determine the psychosociocultural aspects of this difficulty [2].

The doctor–patient relationship is seen as fundamental in the treatment of infertility [3], due to the emotional implications of fertilization procedures. Previous studies have shown that the increased prevalence of mood disorders such as stress, anxiety and depression among infertile women is associated with treatment for infertility. The lack of children, the various difficulties that infertile women go through, uncertainty, socio-emotional pressure, but also repeated treatments are sources of suffering and have emotional implications for women during treatment procedures [4,5,6,7].

The difference between reproductive sciences and other clinical sciences in the context of research, which is the novelty of this study, considers the current global crisis, during which certain internal mechanisms are activated and patients’ needs are more deeply felt and lead to exasperation.

The COVID-19 pandemic may have negatively impacted women’s relationships with their doctors during infertility treatments. For example, we found that 33% of participants stated that the pandemic affected their relationship with their doctor, and 44% discontinued contact with the specialist and medical procedures during this time. Our research is, therefore, important in that it helps to better understand the factors that support effective patient–doctor relationships during this period.

We consider that policy schemes need to be implemented as a way of changing infertility specialists’ behavior, enabling them to better construct and utilize this dyadic relationship. Regarding COVID-19, it could affect how communication influences trust and, therefore, collaboration, because there was a long-term interruption of relationships and even other psychological mechanisms related to confusing personality structures, including insecurity and lack of trust in the medical system.

In building a doctor–patient therapeutic relationship, communication is the central process of transmitting and receiving information [8]. This is important in order to provide high-quality healthcare. A doctor’s communication and interpersonal skills include the ability to gather information; facilitate accurate diagnosis; provide appropriate counseling, therapeutic instructions and excellent patient care with the ultimate goal of achieving the best outcome and satisfaction for the patient, which are essential in healthcare [9,10]. Empathy is an essential key factor in strengthening a relationship, and has significant importance, especially in this medical context. The feeling of being understood and accepted by the doctor is itself a definition of the patient’s perception of the doctor’s empathy. This understanding and acceptance comprise two components: cognitive empathy and affective empathy [11].

The cognitive aspect of empathy is defined as the doctor’s ability to accurately understand the mental state of their patients and effectively communicate this perspective to patients. The affective aspect of doctor empathy is defined as the doctor’s ability to respond and improve the emotional state of their patients [12]. The results of this previously cited study showed that the doctor’s empathic communication skills significantly influenced patient satisfaction and compliance. The emotional aspects of the doctor–patient relationship (for example, collaboration with the doctor and his/her affective empathy) were the most important aspects in increasing satisfaction and compliance for the participants in this study.

In addition, in an effective communication between a doctor and patient, the “mutual understanding” [13] is important, which leads to a relationship based on trust.

Trust is a fundamental aspect of the doctor–patient relationship. Balint believes that as the duration of the doctor–patient relationship increases, the doctor gains the patient’s trust, due to the fact that their knowledge of the patient gradually increases. This allows them to improve their understanding of the patient’s needs, but also their time management skills, so that each consultation is more efficient.

The doctor’s trustworthiness is generally associated with competence, effective communication, care, support, honesty and good collaboration [14]. On the other hand, factors such as a doctor’s poor communication skills, using and not explaining medical terminology, specialists who do not pay attention to patients’ symptoms and inconsistency between doctors’ goals and patients’ expectations create a gap in the doctor–patient relationship [15]. All these factors have an impact on the patient’s confidence level [16,17].

The patient’s trust is related to the concept of satisfaction in the relationship with the doctor, but conceptually different from it.

Satisfaction is another important concept and is often evaluated in doctor–patient relationship research. Although satisfaction refers to the patient’s views on the doctor’s actions, trust refers to the doctor–patient relationship, which is largely based on perceptions of the doctor’s motivations. Confidence also has a strong emotional component, which is not present in satisfaction.

Patients who feel understood and satisfied with the results of treatment are more likely to continue maintaining the relationship with the doctor [18]. The main predictors of satisfaction in the study by Little et al. [19] were patients’ perceptions of communication and collaboration with the doctor, as well as his positive attitude. The same study showed that satisfaction is a predictor of treatment adherence (patient compliance).

Compliance refers to the active involvement and responsibility of patients in the treatment process [20], in which patients closely cooperate with health care providers to maintain the continuity of treatment and ensure good health [21].

Evidence from qualitative studies suggests that patients’ trust is a ‘state’, not a ‘trait’ [14] and, therefore, may change depending on the patient’s situation. This is also the case in the concept of patient satisfaction in their relationship with the infertility specialist. When treatment procedures are successful, both confidence and satisfaction with the doctor will increase. However, there may be situations where the confidence of patients diagnosed with infertility may decline. Many patients are disappointed with healthcare providers, especially after treatment failure. Thus, patients may face difficulties in requesting another medical opinion or in changing their infertility specialist.

Therefore, in order to help those in need, a doctor must know not only scientific aspects and practical skills, but also understand human nature. The patient is not only a group of symptoms. The patient is a human being, who simultaneously feels worried and hopeful and who seeks solutions, support and trust. If the doctor–patient relationship is, by definition, a complex relationship in the treatment of assisted reproduction, there are nuances that require a specific approach.

During the COVID-19 pandemic, the most important Reproductive Medicine Societies advised the cessation of newly started assisted reproductive treatments in order to avoid the strain on the healthcare system. For some patients, the indefinite postponement of assisted reproductive treatments could lead to an irremediable deterioration of reproductive prognosis [22,23,24], which is why we chose to analyze the need to maintain the patient–doctor relationship and the adaptive mechanisms of patients.

This study aimed to synthesize and conceptualize the basic elements of the relationship between the infertility doctor and the patient with infertility and to highlight possible implications for clinical practice.

The general objective of the study was to establish the links between the fundamental concepts that define the doctor–patient relationship: communication, empathy, trust, collaboration, compliance and satisfaction.

The specific objectives, which were derived from the general objective, refer to the establishment of relationships among the following hypotheses:

**Hypothesis** **1.**
*Communication will be related to treatment compliance, mediated by cognitive empathy, trust in the doctor and collaboration.*


**Hypothesis** **2.**
*Communication will be related to treatment compliance, mediated by trust in the doctor and collaboration.*


**Hypothesis** **3.**
*Communication will be associated with treatment compliance, mediated by affective empathy, trust in the doctor and collaboration.*


**Hypothesis** **4.**
*Communication will be related to treatment compliance, mediated by affective empathy, trust in the doctor and patient satisfaction.*


**Hypothesis** **5.**
*Communication will be associated with satisfaction, mediated by affective empathy and trust in the doctor.*


**Hypothesis** **6.**
*Communication will be related directly to trust in the doctor or will be mediated by affective empathy or/and cognitive empathy.*


## 2. Materials and Methods

### 2.1. Procedures and Participants

The study was conducted during the state of alert (May–June 2020) in the context of the COVID-19 pandemic, to better understand the patient’s perception of the relationship with the specialist and how this relationship has an impact on treatment compliance and patient satisfaction.

Participants were recruited through social media ads (support communities set up on Facebook). Data were collected online through the Google Form platform. Before the administration of the questionnaires, participants were informed about the purpose of the study; the use and storage of data; and how the data would be stored. Regarding the inclusion and exclusion criteria, only participants who had contacted a medically assisted human reproduction specialist were invited to participate in the study. Participation in the study was voluntary; all participants agreed to participate in the study.

The present study respects the principles of research ethics regarding the confidentiality of the data collected and ensures the anonymity of the participants. The instruments used and the procedure were noninvasive.

### 2.2. Psychological Assessment

Participants answered a set of data collection questions based on socio-demographic information, such as age, marital status and educational status, and infertility, such as duration, type and cause of infertility, if following any treatment, number of fertilizations/inseminations, the duration of their relationship with their doctor and aspects regarding the choice of doctor and fertilization clinic, as well as the following psychological scales:(a)For the evaluation of doctor–patient communication, we used the Doctor–Patient Communication Questionnaire developed by Sustersic et al. [25]. It contains 13 items, scored on a Likert scale with four response options, from 1 (meaning “No”) to 4 (meaning “Yes”). Examples of questions are as follows: “Was it easy to understand what the doctor told you?” “Did the doctor explain the advantages and disadvantages of the treatment or treatment plan?”.(b)Another instrument used was the Empathy Scale from the study of Kim et al. [12]. It measures eight variables, but in this study, we used only four subscales. Two subscales relate to the patient’s perception of the doctor’s communication skills: cognitive empathy (example item: “This doctor almost always knows exactly what I mean.”) and affective empathy (example item: “This doctor shows that he cares about my psychological well-being.” “I feel comfortable asking this doctor questions about my problem”). The other two subscales involve patient satisfaction (example item: “In general, I am satisfied with this doctor.”) and compliance with treatment (example item: “I followed exactly the treatment schedule prescribed by this doctor.”). Each subscale of the questionnaire comprises 2–7 items measured on a Likert scale from 1 to 5, where 1 means “Strong disagreement” and 5 “Strong agreement”.(c)The Trust in Physician Scale was used to assess the confidence of patients diagnosed with infertility in their specialists [26]. The scale consists of 11 items measured on a Likert scale from 1 (“Strong disagreement”) to 5 (“Strong agreement”). Higher scores indicate a higher level of confidence in the doctor. Examples of items are as follows: “I trust that my doctor puts my medical needs above all other considerations when treating my problems.” “Sometimes I worry that my doctor may not keep the information we are talking about confidential.”

For all the instruments mentioned above, in Table 1, the fidelity indices reported by the authors, in the initial studies and the Cronbach Alpha fidelity indices calculated for the present study, are shown.

### 2.3. Study Design and Statistical Analysis

The present study was a nonexperimental, cross-sectional one; following the method of quantitative analysis, the data were collected using questionnaires. The research plan was specific to the analysis of the pathway with the mediation effect, in which we tested several hypotheses.

To test the proposed hypotheses, analyses were performed in R (R Core Team, Vienna, Austria) [27], and path analysis was used as it allows all variables to be observed. Statistical analysis was performed using the bootstrap method. Since the variables were observable, we used path analysis. We used the maximum probability method. We also used the bootstrap method using a sample of 1000 observations. The effect of confounding variables was not analyzed; the analysis targeted only the variables included in the model. Figure 1 illustrates the diagram used to draw the proposed model.

## 3. Results

### 3.1. Sample Characteristics

Table 2 shows the demographic and clinical variables for all participants and Table 3 emphasize descriptive statistics of the variables included in the explanatory model. The sample consists of 151 women with fertility problems (*n* = 151) between 21 and 46 years old (M = 33.34, SD = 4.63). Regarding the duration of infertility, the participants’ responses varied between 1 and 15 years (M = 5.40, SD = 3.82).

Regarding the number of infertility specialists they had come into contact with, the answers varied between 1 and 10 (M = 2.29, SD = 1.40). Furthermore, 104 of the study participants chose to turn to an infertility specialist based on recommendations (68.9%); 41 participants (27.2%) searched the internet on their own; and 6 participants (4%) stated that they met their doctor by chance. The majority of participants (127–84.1%) stated that they chose their fertilization clinic according to the specialist doctor; 18 participants (11.9%) chose the clinic according to reputation; and 5 participants (4%) chose the clinic according to other characteristics (embryologist and success rates).

Regarding the duration of the doctor–patient relationship, 98% of the participants had had a relationship with their current doctor for a maximum of 4 years, and 2% of the participants had had a relationship of more than 5 years with their current specialist doctor.

Regarding the relationship with the doctor during the COVID-19 pandemic, 33% of the study participants stated that the pandemic affected their relationship with their doctor, and 44% discontinued contact with the specialist and medical procedures between March and May 2020.

### 3.2. Testing the Study Hypothesis

The results of the model fit indicators are shown below.

According to the indicators presented in the Table 4, the data of our study support the proposed model of the doctor–patient relationship. Thus, we found that the CMIN/DF indicator had an acceptable value (CMIN/DF < 3), according to Hu and Bentler [28] and Kline [29]; the same was observed for GFI and CFI indicators, with values close to 1. In terms of the RMSEA value, some authors [30] recommend that models exceeding 0.1 are not implemented, but the model proposed in this study was at the limit of this threshold (RMSEA = 0.09), and the lower indicator (0.081) had a recommended value for model support [28,29]. Other information extracted from the results of the processing of the proposed model is presented in the following tables, which represent the relationships among the study variables, and between predictors and mediators, within the proposed explanatory model.

In Table 5 and Table 6 we analized the relationship between the study variables. No hypothesis was confirmed. The only significant finding is that affective empathy mediates the relationship between communication and trust in the doctor.

## 4. Discussion

### 4.1. Results of the Study in the Context of What Is Known

The doctor–patient relationship is a major component of the health care process. A good doctor–patient relationship can strengthen patients’ self-confidence, motivation and positive perception of their health, which can influence their treatment outcomes [31].

The findings of this study show that the effective use of empathic communication skills can be one of the best ways to improve a patient’s trust in their doctor. This result is significant because this type of affective empathy, which infertile patients may perceive in communication with the specialist, facilitates the construction of a professional relationship based on emotional connection. For doctors to show affective empathy, they need to have the ability to listen carefully to the experience of the infertile person/couple, to understand how they feel and why they feel that way. Additionally, doctors who have affective empathic skills reflect on patients’ feelings and how they can relate to them, showing a high degree of compassion. These doctors can gain a more complete picture of the patients’ situation and emotionally support them. Once patients perceive this connection with their doctors, they will have more trust in them and in the information they provide, as well as in the quality of the medical act.

Thus, as a single management strategy, fertilization clinics and hospitals that have infertility treatment wards could adopt a training program to increase the empathic skills of specialists.

Given that previous studies are mainly reviews that conceptualize the psychological or medical constructs involved in the infertility specialist–patient relationship [32], observing this through a detailed search and the aspects that come from the clinical area, we consider the results of the present research valuable.

### 4.2. Limitations of the Study

This study has several limits. From a methodological point of view, an important limitation is the transversal design. For this reason, no causal inferences can be made about the relationships between variables. Additional experimental research on other and extended infertile populations is needed to examine the effects of communication on patient satisfaction and compliance. Patients’ communication style may also influence the doctor–patient relationship, but this was not included in this study. Additionally, this study did not include patients’ expectations regarding the doctor’s communication styles. Some patients may prefer authoritarian doctors, while others may not [33]. Another methodological limitation refers to the use of general assessment tools, which are not built to be used in the context of infertility.

This study may be limited due to the application of self-assessment scales and the convenience of the research groups. Thus, there is a risk that the data reported by participants will be affected by their capacity for self-knowledge and their tendency towards social desirability. Due to the voluntary participation of patients with infertility in this study, the generalization of the results is limited.

### 4.3. Strengths and Clinical Implications of the Study

Regarding the strengths of the present study, the main value is that this is the first study to analyze the relationship between an infertility specialist and a patient, thus bridging the gap in the medical and psychological literature in the field of infertility.

Additionally, this study presents very valuable theoretical, applied, methodological and clinical contributions.

The main theoretical contributions are the identification and operationalization of the conceptual dimensions that create the relationship between an infertility specialist and a patient, namely, communication, empathy, trust, satisfaction and compliance.

Specifically, the real application and clinical challenges are the development and activation of these qualities and abilities in the medical professional environment. This study has practical implications for medical teams specialized in assisted human reproduction. Thus, it is necessary for them to participate in programs for the continuous training of empathic communication skills, oriented in the sense of collaboration with the patient, given the sensitivity of this case, but also to guide patients in psychological assessment and participation in therapeutic support groups.

In the scientific literature, there are few evidence-based studies regarding the doctor–patient relationship. Most descriptions of the basic dimensions of this relationship are primarily derived from conceptual analysis and not from empirical research, which constitutes a gap in the literature. As an example, Emanuel and Dubler [32] suggested that the ideal doctor–patient relationship consists of choice, competence, communication, compassion and continuity.

## 5. Conclusions

This paper emphasizes the importance of basic concepts in the doctor–infertile patient relationship, such as empathy, communication, trust, compliance and the feeling of satisfaction. For study participants diagnosed with infertility, communication and empathy are important factors in increasing trust in the infertility specialist. By increasing patient trust, it is possible for the doctor to have a beneficial influence on the patient’s health and to achieve the desired results. In conclusion, the doctor–patient relationship can be seen as more of an art than a science. No measuring instrument can capture all the nuances of this complex relationship. However, the challenge is to make these concepts operational.

## Figures and Tables

**Figure 1 healthcare-09-01649-f001:**
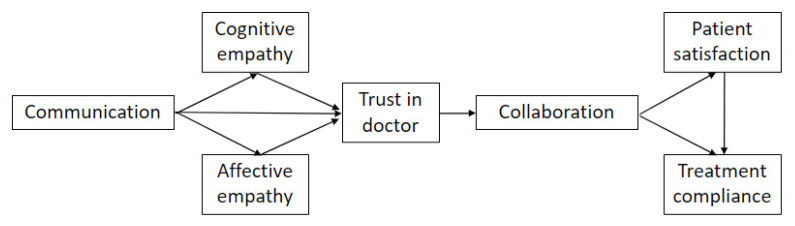
Diagram of the explanatory model regarding the doctor–patient relationship.

**Table 1 healthcare-09-01649-t001:** Fidelity indices for the instruments used.

Instruments	Cronbach Alpha Reported by the Authors	Cronbach Alpha for the Present Study
Doctor–patient communication questionnaire	0.89	0.96 (0.95–0.97)
Empathy Scale—Cognitive empathy	0.68	0.88 (0.84–0.91)
Empathy Scale—Affective empathy	0.87	0.90 (0.87–0.92)
Empathy Scale—Collaboration	-	0.93 (0.91–0.94)
Empathy Scale—Treatment compliance	0.78	0.91 (0.87–0.93)
Empathy Scale—Patient satisfaction	0.87	0.91 (0.89–0.93)
Trust in Physician Scale	0.82	0.90 (0.87–0.92)

**Table 2 healthcare-09-01649-t002:** Demographic and clinical variables for all the participants (*n* = 151).

Demographic and Clinical Variables	Frequency	Percentage
Educational status		
General school	5	3.3%
High school	21	13.9%
Post-secondary school	16	10.6%
Undergraduate studies	50	33.1%
Master’s Degree	54	35.8%
Doctoral studies	5	3.3%
Marital status		
Married	132	87.4%
I live with a partner	16	10.6%
I have no partner	3	2%
Environmental status		
Urban	121	80.1%
Rural	30	19.9%
The type of infertility		
Primary	109	72.2%
Secondary	42	27.8%
The cause of infertility		
Female causes	66	43.7%
Male causes	15	9.9%
Both	32	21.2%
Idiopathic/Inexplicable	38	25.2%
Treatment/fertilization procedures		
Yes, I did at least one In Vitro Fertilization (IVF)	53	35.1%
Yes, I did at least one artificial insemination (AI)	15	9.9%
Both (both IVF and AI)	28	18.5%
I’m at the beginning of treatment	48	31.8%
Not yet	7	4.6%
Repeated treatment	69	45.7%
Yes, I went through several treatment procedures		
No, only one treatment so far	37	24.5%
No treatment procedure so far	45	29.8%
Patient under observation at:		
Private clinic in the country	121	80.1%
State hospital in the country	17	11.3%
Private clinic abroad	13	8.6%
Treatment stage		
Preliminary stage: Analyses, ultrasounds	54	35.8
Ovarian stimulation treatment	27	17.9
Oocyte harvesting	4	2.6
Embryo transfer	17	11.3
Waiting for the result of the pregnancy test	4	2.6
Pregnancy in progress	16	10.6
Negative result of embryo transfer	7	4.6
I gave birth	7	4.6
Treatment break	15	9.9

**Table 3 healthcare-09-01649-t003:** Descriptive statistics of the variables included in the explanatory model.

Variable	M	SD
Treatment compliance	9.53	1.16
Communication	46.81	8.58
Cognitive empathy	11.91	2.87
Affective empathy	28.34	6.19
Trust in doctor	45.15	8.98
Collaboration	21.39	4.52
Patient satisfaction	9.53	1.16

**Table 4 healthcare-09-01649-t004:** Matching indicators obtained for the tested model.

Indicators	Value
CMIN/DF (Minimum discrepancy/Degree of Freedom)	2.12
GFI (The Goodness-of-Fit Index)	0.624
CFI (The Comparative Fit Index)	0.867
RMSEA (min/max) (root mean square error of approximation)	0.087 (0.081/0.092)
NFI (The Normed Fit Index)	0.777
RFI (The Relative Fit Index)	0.763
IFI (The Incremental Index of Fit)	0.868
TLI (The Tucker–Lewis Index)	0.859

**Table 5 healthcare-09-01649-t005:** The relationship between the study variables.

Variables	*p*	Variables	Size Effect	Z Score	SE	95% CI
Trust in doctor	<---	Cognitive empathy	0.146	1.165	0.388	−0.356–1.121
Trust in doctor	<---	Affective empathy	0.503 ***	4.181	0.172	0.397–1.061
Trust in doctor	<---	Communication	0.262 *	2.536	0.107	0.059–0.478
Collaboration	<---	Trust in doctor	0.818 ***	0.000	0.030	0.349–0.467
Satisfaction	<---	Collaboration	0.881 ***	0.000	0.039	0.650–0.805
Compliance	<---	Satisfaction	−0.180	0.298	0.054	−0.158–0.057
Compliance	<---	Collaboration	0.508 **	0.045	0.065	0.650–0.805
Cognitive empathy	<-->	Communication	0.789 ***	0.000	0.025	0.212–0.308
Affective empathy	<-->	Communication	0.856 ***	0.000	0.029	0.567–0.0681

Note: * *p* < 0.05, ** *p* < 0.01, *** *p* < 0.001. Arrows: if an arrow goes from x to y means that x has an effect on y.

**Table 6 healthcare-09-01649-t006:** Mediation path and mediation effect regarding the study variables.

Mediation Path	Mediation Effect
Communication → Cognitive Empathy → Trust in doctor → Collaboration → Treatment Compliance	0.048
Communication → Trust in doctor → Collaboration → Treatment Compliance	0.109
Communication → Affective Empathy → Trust in doctor → Collaboration → Treatment Compliance	0.179
Communication → Cognitive Empathy → Trust in doctor → Collaboration → Patient Satisfaction→ Treatment Compliance	−0.015
Communication → Affective Empathy → Trust in doctor → Collaboration → Patient Satisfaction → Treatment Compliance	−0.056
Communication → Cognitive Empathy → Trust in doctor	0.115
Communication→ Affective Empathy → Trust in doctor	0.430 ***

Note: *** *p* < 0.001.

## Data Availability

The datasets generated during and/or analyzed during the current study are available from the corresponding author on reasonable request.

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
