# Peer review of "The Relationship between the Infertility Specialist and the Patient during the COVID-19 Pandemic"

_healthcare, 2021, doi:10.3390/healthcare9121649_

Round 1

Reviewer 1 Report

Thank you very much for the opportunity to review the manuscript entitled “the relationship between the infertility specialist and the patient during the COVID-19 pandemic”. There is no doubt that the doctor-patient relationship plays a critical role in patient service utilization and adherence, and there is still much unknown within this topic. However, I have some major concerns about this study's novelty, objectives, research design, and result reporting. My comments are as follows:

  1. The authors aimed to investigate the impact of communication on patient’s compliance and the mechanisms of action. The introduction listed several key mediators such as empathy, trust, and satisfaction and indicated their relationship with communication. However, as the authors stated, it seems that the positive associations of communication with these variables have been widely reported in healthcare settings. In this case, what is the novelty of conducting a similar study in reproductive sciences? In other words, what are the difference between reproductive sciences and other clinical sciences in terms of the research context? Readers may clearly appreciate the novelty of this study once this question is resolved in the introduction section.
  2. The title mentioned the COVID-19 pandemic. It seems that it is a particular context for this study. However, it was not very clear whether the authors intended to measure the impact of the COVID-19 pandemic on the doctor-patient relationship or, in theory, the impact of communication on patient compliance would change during the COVID-19 pandemic? I feel confused as COVID-19 was mentioned in the title but not in the research objectives.
  3. I suggest moving the hypotheses to the introduction section and adding a figure clearly presenting the hypothesized model of the relationships across these variables.
  4. In terms of procedure and participants, did this study only include those who had ever contacted an infertility specialist? Please provide more details on the participants in terms of the inclusion and exclusion criteria.
  5. The authors reported statistical analysis in the results section rather than the methods section. I suggest the authors remove this part to the end of the methods section and provide the statistical strategy in more detail. For instance, whether the confounding factors were controlled in the SEM; have the results been adjusted using the bootstrapping technique?
  6. The present study seemed to treat the variables as measurable variables and used scales to measure rather than treating them as latent variables. In this case, I suggest replacing the ellipses with rectangles in Figure 1 and presenting the distribution of values of these variables in a new table after Table 2.
  7. In terms of the goodness of fit, the present manuscript reported GFI=0.624, CFI=0.867, NFI=0.777, RMSEA=0.087 (Table 3). And the authors stated that GFI and CFI values were close to 1 (p7, lines 232-233). As far as Table 3 presented, GFI was significantly below 1. Moreover, extant literature suggests a cut-off criterion of >0.9 for NFI GFI and CFI, and a cut-off criterion of <0.08 or <0.05 for SRMR and RMSEA (Schermelleh-Engel K, Moosbrugger H, Müller Evaluating the Fit of Structural Equation Models: Tests of Significance and Descriptive Goodness-of-Fit Measures. Methods of Psychological Research Online, 2003, 8(8): 23-74). The low goodness of fit may compromise the validity of the model, and therefore, the results.
  8. Table 5: the authors generally reported the path and mediation effect whether they were significant, rather than reporting the specific result of each path and the total effect size of the model. I suggest the authors present the effect size and p-value for each path and the total effect. This can allow readers to have a better idea of the total effect of communication on compliance and how much was mediated by the mediators.
  9. The authors did not compare the findings with those from prior studies in the discussion section. This may limit their contribution to knowledge, and therefore, the novelty.

Author Response

Response to editor and reviewers' comments

Open Review

Thank you very much for the opportunity to review the manuscript entitled “the relationship between the infertility specialist and the patient during the COVID-19 pandemic”. There is no doubt that the doctor-patient relationship plays a critical role in patient service utilization and adherence, and there is still much unknown within this topic. However, I have some major concerns about this study's novelty, objectives, research design, and result reporting. My comments are as follows:

Thank you very much for your appreciation of our study.

See below the author's response.

Reviewer 2 Report

I think that this piece of work will be of interest to researchers and practitioners working in infertility and related healthcare. However, some improvements are needed before the manuscript can be considered as suitable for publication. I have listed my specific comments below, but overall, I would recommend making the specific aims, purpose, and rationale of the study clearer.

Specific comments:

The abstract needs to be more concise and to the point. Tell the reader what you did, why and the key findings/conclusions. Details such as the specific , CMIN / DF, RMSEA,  NFI, CFI, RFI, IFI and TLI are unnecessary.

More importantly, the abstract does not actually tell the reader what the study is about currently! Phrases such as “information related to infertility, as well as information on the relationship of patients 20 with the infertility specialists” and “The study draws attention to the importance of basic concepts in the relationship of 26 infertility specialists with infertile patients” are too vague. What were the more specific hypotheses that you were testing? There is no point in telling the reader the results of the SEM if they do not actually know what was tested within the model.

On page 2:

“The doctor-patient relationship is seen as fundamental in the treatment of infertility 45 [3], due to the emotional implications of fertilization procedures [4].” More detail needed here. What are the emotional implications of fertilization procedures?

“… collaboration with the doctor and his affective empathy…” I would use his/her affective empathy.

“The doctor’s trustworthy is…” should be ‘The doctor’s trustworthiness is…”

On page 3:

“During the COVID-19, the most important Reproductive Medicine Societies advised 109 to stop the start of new Assisted Reproductive Treatments in order to avoid the strain on 110 healthcare system. For some patients the indefinite postponement of Assisted Reproductive Treatments could lead to an irremediable deterioration of reproductive prognosis [19- 112 21].”  It was not clear to me how this statement about the postponement of treatment due to COVID-19 was related to your paper?

I think that the specific hypotheses, which you currently include in the Materials and Methods section, would be better placed at the end of the introduction. I.e. replace the list that you currently have under this sentence “The specific objectives, which derive from the general objective, refer to the establishment of the relations between: …” with the specific hypotheses.

On page 6:

“The proposed explanatory model was one in which a number of variables were influenced, together contributing to the explanation of the doctor-patient relationship.” This sentence does not make sense.

On page 8:

“*** The relationship is highly significant at a significance level of .1%” By this do you mean that p < .01? I think it would make more sense to most audiences to write what the p value was less than rather than saying ‘a significance level of .1%’.

I do not like how you suddenly list results using a), b), c), d) etc. At first, I thought these would correspond with the content of Table 5 but they do not. So why are some results reported in the table and some listed in the text? It is confusing for the reader.

Relatedly, it would be helpful for you to indicate what degree of significance the paths and mediating effects have (p < .05? p < .01? p < .001?) and their effect size.

On page 9:

“according to the statistical indicators mentioned above, ie CMIN / DF = 261 2.124, RMSEA = .087 and the p value of the Chi-square was less than 5% significance level, 262 the model was accepted.” I think that when conducting structural equation modelling, you want the chi-square to not be significant (but it often is, especially when using larger samples).

Are you using structural equation modelling (SEM) or a path analysis? SEM implies that you are modelling your variables as latent, rather than observed, but it is not clear from your manuscript whether you have done this or not. Just make sure that you make it clear which type of analysis you are adopting.

It would benefit your results section to refer back to your specific hypotheses and highlight whether these were supported or not. E.g. through the use of statements such as “… thus supporting Hypothesis 1” or “… (H1 supported).”

You call section 4.1 “4.1. Results of the study in the context of what is know.” Firstly, I assume you mean ‘known’ not ‘know’. Secondly, in this section you do not really refer back to existing literature. You just summarise the main findings. To improve, talk about the findings from previous research more and how they support/conflict with your own findings.

“Additional experimental research on older populations is needed to examine the effects of communication on patient satisfaction and compliance.” Why older populations in particular?

On page 10:

“The methodological contributions are given by the analyzes performed during the coronavirus pandemic, in which the physical contact was more and more diminished, following in this context the doctor-patient relationship. In the scientific literature, there are few evidence-based standards regarding the doctor-patient relationship.” Sorry, I still do not understand how your study or methods are related to the covid-19 pandemic. You need to make this clear to the reader or remove reference to covid-19 in the manuscript.

Author Response

Response to editor and reviewers' comments 

I think that this piece of work will be of interest to researchers and practitioners working in infertility and related healthcare. However, some improvements are needed before the manuscript can be considered as suitable for publication. I have listed my specific comments below, but overall, I would recommend making the specific aims, purpose, and rationale of the study clearer.

Thank you very much for your appreciation of our study.

See below the author's response.

Round 2

Reviewer 1 Report

I appreciate the authors' reply. The authors have addressed some of my comments. However, some of my major concerns have not been fully addressed:

  1. I raised the question about the novelty and asked for the specific differences between reproductive sciences and other clinical sciences. The authors just mentioned, "Thank you for your suggestions. We agree with you and we added that we believe that the difference between reproductive sciences and other clinical sciences in the context of research and that is the novelty of this study is given by the time of evaluation, a time that is a global crisis, through which certain internal mechanisms are activated and patients' needs are more deeply felt and sometimes with exasperation transmitted. We consider it a novelty the usefulness to point better in this context, the stringent psychological mechanisms which generate the relationship with the doctor during the COVID-19 pandemic, because 33% of the study participants stated that the pandemic affected the relationship with the doctor, and 44% discontinued contact with the specialist and medical procedures, which is the need to know the evolution of these key mediators". The authors only attributed the novelty to the difference between reproductive sciences and other clinical sciences without specifying the exact differences, which left my question not answered, what are the unique chrematistics of reproductive sciences that make it important to investigate this specific area? The other question is that did or could COVID-19 affect how communication influences trust and, therefore the collaboration? If it did/could, please specify the potential mechanism.
  2. In terms of the goodness of fit, my major concern was the GFI = 0.624, as previous studies suggested a GFI >0.9.
  3. Regarding Table 5. Please provide more information such as z score, SE, and 95% CI.
  4. Methods: Which estimation method was adopted to assess the goodness of fit? Did the authors include confounding variables in the path analysis? What type of bootstrapping methods was employed? Parametric or non-parametric? What is the bootstrapping sample?
  5. Table 2: Some categories have a very small sample size (Vocational school n = 2; I have no partner n = 3). I suggest the authors re-classify them.
  6. Discussion: I suggest the authors carry out a more in-depth discussion of their findings, for instance, why affective empathy is the only mediator of the relationship between communication and trust?

Author Response

Response to editor and reviewers' comments

Open Review

I appreciate the authors' reply. The authors have addressed some of my comments. However, some of my major concerns have not been fully addressed:

  1. I raised the question about the novelty and asked for the specific differences between reproductive sciences and other clinical sciences. The authors just mentioned, "Thank you for your suggestions. We agree with you and we added that we believe that the difference between reproductive sciences and other clinical sciences in the context of research and that is the novelty of this study is given by the time of evaluation, a time that is a global crisis, through which certain internal mechanisms are activated and patients' needs are more deeply felt and sometimes with exasperation transmitted. We consider it a novelty the usefulness to point better in this context, the stringent psychological mechanisms which generate the relationship with the doctor during the COVID-19 pandemic, because 33% of the study participants stated that the pandemic affected the relationship with the doctor, and 44% discontinued contact with the specialist and medical procedures, which is the need to know the evolution of these key mediators". The authors only attributed the novelty to the difference between reproductive sciences and other clinical sciences without specifying the exact differences, which left my question not answered, what are the unique chrematistics of reproductive sciences that make it important to investigate this specific area? The other question is that did or could COVID-19 affect how communication influences trust and, therefore the collaboration? If it did/could, please specify the potential mechanism.

Thank you for your kind comments regarding our paper. We agree with you and we added that

The difference between reproductive sciences and other clinical sciences in the context of research and that is the novelty of this study is given by the assessmant that is a global crisis time, through which certain internal mechanisms are activated and patients' needs are more deeply felt and sometimes with exasperation transmitted.

The COVID-19 pandemic may have negatively impacted women’s relationships with their doctors during infertility treatments. For example, we found that 33% of participants stated that the pandemic affected the relationship with the doctor, and 44% discontinued contact with the specialist and medical procedures during this time. Our research is therefore important in that it helps to better understand the factors that support effective patient-doctor relationships during this period.

We consider that policy schemes need to be implemented as a way of changing infertility specialists’ behavior, forcing them to better construct and utilize this dyadic relationship.  As regarding COVID-19, it could affect how communication influences trust and, therefore the collaboration, because there was a long-term interruption of the rela-tionship and not only that, but even other psychological mechanisms also related to confusing personality structures, including insecurity and lack of trust in the medical system.

  1. In terms of the goodness of fit, my major concern was the GFI = 0.624, as previous studies suggested a GFI >0.9.

Thank you for your comment. Unfortunately we cannot get better indices without changing the model and we do not know if you allow us to do so.

  1. Regarding Table 5. Please provide more information such as z score, SE, and 95% CI.

Thank you for your comment. Attached below is the table of results and you can see the value of z and the confidence interval (ci.lower and ci.upper). The effect reported in the manuscript is on the std.all column. For example, for the relationship trust - cognitive empathy effect size is 0.146, z = 1.165, ci 95% -0.356 - 1.121.

Variables

P

Variables

Size effect

Z score

SE

95% CI

Trust in doctor

<---

Cognitive empathy

0.146

1.165

0.388

-0.356-1.121

Trust in doctor

<---

Affective empathy

0.503***

4.181

0.172

0.397-1.061

Trust in doctor

<---

Communication

0.262*

2.536

0.107

0.059-0.478

Collaboration

<---

Trust in doctor

0.818***

0.000

0.030

0.349-0.467

CSatisfaction

<---

Collaboration

0.881***

0.000

0.039

0.650-0.805

Compliance

<---

Satisfaction

-0.180

0.298

0.054

-0.158-0.057

Compliance

<---

Collaboration

0.508**

0.045

0.065

0.650-0.805

Cognitive empathy

<-->

Communication

0.789***

0.000

0.025

0.212-0.308

Affective empathy

<-->

Communication

0.856***

0.000

0.029

0.567-.0681

  1. Methods: Which estimation method was adopted to assess the goodness of fit? Did the authors include confounding variables in the path analysis? What type of bootstrapping methods was employed? Parametric or non-parametric? What is the bootstrapping sample?

Thank you for your comment. We used the maximum probability method. We also used the bootstrap method using a sample of 1000 observations. The effect of confounding variables was not analyzed, the analysis targeting only the variables included in the model.

  1. Table 2: Some categories have a very small sample size (Vocational school n = 2; I have no partner n = 3). I suggest the authors re-classify them.

Thank you for your suggestions. We agree with you and we reclassified them in this way: vocational school is similar to high school, so we joined the two categories. Regarding the other requirement, we do not know how we could reclassify the fact that 3 participants are single.

  1. Discussion: I suggest the authors carry out a more in-depth discussion of their findings, for instance, why affective empathy is the only mediator of the relationship between communication and trust?

Thank you for your suggestions. We agree with you and we added that: "This result is significant because this type of affective empathy, which infertile patients may perceive in communication with the specialist, facilitates the construction of a professional relationship based on emotional connection. For doctors to show affective empathy, they need to have the ability to listen carefully to the experience of the infertile person / couple, to understand how they feel and why they feel that way. Also, doctors who have affective empathic skills reflect on patients' feelings and how they can relate to them, showing a high degree of compassion. These doctors have a more complete picture of the patients' situation and emotionally support them. Once patients perceive this connection relationship with their doctors, they will have more trust in their doctors and in the information provided by them, as well as in the quality of the medical act."

Reviewer 2 Report

Thank you for allowing me to review this manuscript again. I still have some issues with the manuscript in its current form. These mainly revolve around needing to be clear about exactly what you want to test, how and why. Some improvements to the general writing style would also be helpful as, at times, I could not understand the points you were trying to make in your sentences. 

Specific comments:

In the abstract you say “the only significant mediation effect…” but there has not been any mention of mediation or why it has been implemented in this study in the abstract. To improve, I would just state that “findings demonstrate that affective empathy…”

The new content on lines 43-48 still does not make it clear why the patient-doctor relationship is therefore important. Is it because this can help to offset some of the negative emotional experiences that occur during infertility treatments?

Lines 49-52 do not make sense to me.

Lines 54-58, I can see the point you are trying to make here, but it needs to be presented more clearly. E.g. it could be written as

“The COVID-19 pandemic may have negatively impacted women’s relationships with their doctors during infertility treatments. For example, we found that 33% of participants stated that the pandemic affected the relationship with the doctor, and 44% discontinued contact with the specialist and medical procedures during this time. Our research is therefore important in that it helps to better understand the factors that support effective patient-doctor relationships during this period.”

Lines 132-134, so do you have two sets of data (before and during the pandemic) which you are comparing? These sentences seem to suggest that you do but I thought that you were only testing during covid???

Lines 137-148, I do not like how the hypotheses are in past tense. I think they would be better in present or future tense e.g. “Communication will be related to treatment compliance…”

Line 210 you say you are using structural equations, but in your responses to my previous comments you say you are running a path analysis? It needs to be clear whether you are using a path model or SEM.

The inclusion of Figure 1 is good.  

Table 6, how did you test the mediation effects and their significance? Was this a bootstrap procedure? It should be noted in the manuscript.  

Author Response

Response to editor and reviewers' comments

Open Review

Thank you for allowing me to review this manuscript again. I still have some issues with the manuscript in its current form. These mainly revolve around needing to be clear about exactly what you want to test, how and why. Some improvements to the general writing style would also be helpful as, at times, I could not understand the points you were trying to make in your sentences. 

Specific comments:

In the abstract you say “the only significant mediation effect…” but there has not been any mention of mediation or why it has been implemented in this study in the abstract. To improve, I would just state that “findings demonstrate that affective empathy…”

Thank you for your comments regarding our paper. We agree with you and we have modified the study abstract according to your requirements: "Findings demonstrate that affective empathy mediates the relationship between communication and trust in the doctor."

The new content on lines 43-48 still does not make it clear why the patient-doctor relationship is therefore important. Is it because this can help to offset some of the negative emotional experiences that occur during infertility treatments?

Thank you for your comment. We agree with you and and to clarify this relationship we have added additional information in the discussion: "This result is significant because this type of affective empathy, which infertile patients may perceive in communication with the specialist, facilitates the construction of a professional relationship based on emotional connection. For doctors to show affective empathy, they need to have the ability to listen carefully to the experience of the infertile person / couple, to understand how they feel and why they feel that way. Also, doctors who have affective empathic skills reflect on patients' feelings and how they can relate to them, showing a high degree of compassion. These doctors have a more complete picture of the patients' situation and emotionally support them. Once patients perceive this connection relationship with their doctors, they will have more trust in their doctors and in the information provided by them, as well as in the quality of the medical act."

Lines 49-52 do not make sense to me.

Thank you for your observation. We agree with you and we reformulated the paragraph as follows: "The difference between reproductive sciences and other clinical sciences in the context of research and that is the novelty of this study is given by the assessment that is a global crisis time, through which certain internal mechanisms are activated and patients' needs are more deeply felt and sometimes with exasperation transmitted.

We consider that policy schemes need to be implemented as a way of changing infertility specialists’ behavior, forcing them to better construct and utilize this dyadic relationship.  As regarding COVID-19, it could affect how communication influences trust and, therefore the collaboration, because there was a long-term interruption of the rela-tionship and not only that, but even other psychological mechanisms also related to confusing personality structures, including insecurity and lack of trust in the medical system. "

Lines 54-58, I can see the point you are trying to make here, but it needs to be presented more clearly. E.g. it could be written as

The COVID-19 pandemic may have negatively impacted women’s relationships with their doctors during infertility treatments. For example, we found that 33% of participants stated that the pandemic affected the relationship with the doctor, and 44% discontinued contact with the specialist and medical procedures during this time. Our research is therefore important in that it helps to better understand the factors that support effective patient-doctor relationships during this period.

Thank you for your comment. We agree with you and we modified the paragraph with your suggestion.

Lines 132-134, so do you have two sets of data (before and during the pandemic) which you are comparing? These sentences seem to suggest that you do but I thought that you were only testing during covid???

Thank you for your comment. We agree that this phrase is confusing and we decided to remove it. We do not have two sets of data, the participants were evaluated only once during the pandemic.

Lines 137-148, I do not like how the hypotheses are in past tense. I think they would be better in present or future tense e.g. “Communication will be related to treatment compliance…”

Thank you for your comment and suggestion.  We agree with you and we reformulated the hypotheses using future tense.

Line 210 you say you are using structural equations, but in your responses to my previous comments you say you are running a path analysis? It needs to be clear whether you are using a path model or SEM.

 Thank you for your comment. We agree with your observation and we have modified it in the text as well. As long as all variables are observed it is about path analysis.

Table 6, how did you test the mediation effects and their significance? Was this a bootstrap procedure? It should be noted in the manuscript.  

Thank you for your comment. We used the maximum probability method. We also used the bootstrap method using a sample of 1000 observations. The effect of confounding variables was not analyzed, the analysis targeting only the variables included in the model.

The inclusion of Figure 1 is good.

 Thank you for your comment.
